

# Evaluation of sugarcane promising clones based on the morphophysiological traits developed from fuzz

Bilal Saleem[1], Muhammad Uzair[1], Muhammad Noman[1,2], Kotb A. Attia[3], Muqing Zhang[4], Mona S. Alwahaibi[5], Nageen Zahra[1], Muhammad Kashif Naeem[1], Arif A. Mohammed[3], Sajid Fiaz[6], Itoh Kimiko[7] and Muhammad Ramzan Khan[1]

[1] Functional Genomics and Bioinformatics Lab, National Institute for Genomics and Advanced Biotechnology (NIGAB), National Agricultural Research Centre, Islamabad, Pakistan
[2] Plant Molecular Physiology Laboratory, Department of Biology, Federal University of Lavras, Lavras, Minas Gerais, Brazil
[3] Department of Biochemistry, College of Science, King Saud University, Riyadh, Saudi Arabia
[4] Guangxi Key Lab for Sugarcane Biology, State Key Lab for Conservation and Utilization of Subtropical Agri-Biological Resources, College of Agriculture, Guangxi University, Nanning, China
[5] Department of Botany and Microbiology, College of Science, King Saud University, Al-Riyadh, Saudi Arabia
[6] Plant Breeding and Genetics, The University of Haripur, Haripur, Pakistan
[7] Institute of Science and Technology, Niigata University, Niigata, Japan

Corresponding authors
Kotb A. Attia, kattia1.c@ksu.edu.sa
Muhammad Ramzan Khan,
mrkhan@parc.gov.pk

## ABSTRACT

Sugarcane is one of the critical commercial crops and principal sources of ethanol and sugar worldwide. Unfavorable conditions and poor seed setting rates hinder variety development in sugarcane. Countries like Pakistan directly import fuzz (true seed) and other propagation material from the USA, China, Brazil, *etc.* In this study, we imported fuzz from China, developed 29 genotypes germinating in the glasshouse, and evaluated at field conditions along with two local checks (CPF-251 and HSF-240). Morphophysiological data were recorded, including plant height (PH), cane length (CL), internodal length (IL), tiller number (TN), brix percentage (B), cane diameter (CD), chlorophyll *a* (Chl. *a*), chlorophyll *b* (Chl. *b*), and total chlorophyll (T. Chl). Results showed highly significant ($p < 0.001$) differences among the sugarcane accessions for all the studied traits. High broad-sense heritability (81.89% to 99.91%) was recorded for all the studied parameters. Genetic Advance (GA) ranges from 4.6% to 65.32%. The highest GA was observed for PH (65.32%), followed by CL (63.28%). Chlorophyll leaching assay was also performed at different time points (0, 50, 100, 150, and 200 min). All the genotypes showed the same leaching trend at all times, and better performing genotypes showed less leaching compared to poor performing, indicating the high amount of cutin and wax on the leaf surface. Correlation analysis showed that PH, CL, IL, and TN had significant associations. Principal components analysis (PCA) further confirms these results. Based on PCA and correlation results, PH, CL, IL, and TN can be utilized as a selection criterion for sugarcane improvement. Genotypes such as NS-4a, NS-5, NS-6, NS-8, NS-9, and NS-15 are recommended for future breeding programs related to sugarcane variety development.

## INTRODUCTION

Sugarcane (*Saccharum officinarum*) is a cash crop that serves as a source of food and biofuel and fulfills global energy demand (*Hess et al., 2016*; *Noman et al., 2022*). The modern sugarcane (*Saccharum* spp., Poaceae) crop has an economic value of US$90 billion and provides 80% of the world's sugar and 40% of ethanol (*Zhang et al., 2022*). In Pakistan, the agriculture sector occupies 22.7% of the GDP, of which 0.8% belongs to sugarcane (*Enonymous, 2022*). Pakistan ranks fourth in global sugarcane production (81 M tons) after China (108.6 M tons), India (370.5 M tons), and Brazil (757.1 M tons). However, its sugarcane yield (695.3 M hg/ha) is lower compared to the area harvested (1,165 M ha) ranking it fifth worldwide. The USA has the least harvested area (383.5 M ha) and the highest yield (854 M hg/ha) among all sugarcane-producing countries. Therefore, developing high-yielding sugarcane varieties is required to tackle the ever-rising problem of food insecurity.

Sugarcane genetics are now recognized to be among the most complicated in the plant kingdom. Sugarcane is a complex octoploid crop, and its exact genetics are still not fully understood, but research has revealed some important aspects of its genome. Modern sugarcane genotypes have evolved from interspecific hybridization between disease-resistant *S. spontaneum* and high-sucrose containing *S. officinarum*. This evolution of sugarcane genotypes was driven by polyploidization, and up to 70% of the flowering species originated shortly after this phenomenon (*Sica, 2021*), which resulted in chromosome numbers up to 100–130 distributed among ∼12 homologous groups and the total genome size reaching 10 Gb (*Thirugnanasambandam, Hoang & Henry, 2018*). The complex sugarcane genome poses constrains in the development of high-yielding, disease-resistant, and higher sucrose and fiber content varieties. Traditional sugarcane breeding methods involved backcrossing to develop genotypes (*Sica, 2021*). The recent improvement in the sugarcane genome assembly has led to the development of SNP chip arrays like 45K, 345K, and 100K (*Aitken et al., 2016*; *McNeil et al., 2017*; *You et al., 2019*), which can help breeders develop improved genotypes through genomic selection and genome-based breeding. It will also promote the protection of Intellectual Property Rights through high-throughput fingerprinting of approved sugarcane varieties (*Saleem et al., 2022*).

Approximately 60 sugarcane varieties are approved in Pakistan, most of which are selected from the exotic fuzz (true seed) of different sources, such as Canal Point, Florida (CP) and Houma, Louisiana, USA (Ho), BSES (Q) and CSIRO, Australia (CS), SASRI, South Africa (N), Copersucar (SP) and CanaVialis, Brazil (CV), MSIRI, Mauritius (M), and SRI, Sri Lanka (SL) (*Afghan et al., 2013*). The development of new varieties has been hindered by unfavorable conditions of temperature (30–42 °C), humidity (35–50%), required day length during the flowering period in most parts of the growing areas, and the limited capacity of sugarcane breeding programs. Although flowering occurs in Thatta (24.7475°N, 67.9106°E), a region of Sindh province (with temperature range of 20–32 °C and humidity of 50–60% during the flowering time) in the southern part of the country near the coastal line, a lack of breeding facilities limits the development of varieties. Therefore, Pakistan relies on imported germplasm and selects potential clones to release as varieties

in the country. In 2009, the National Sugar and Tropical Horticulture Research Institute (NSTHRI) in Thatta evaluated the sugarcane varieties based on different parameters, such as cane and sugar yield, of 22 genotypes developed from the exotic fuzz of US origin and compared them with two local check varieties (*Junejo et al., 2009*). The Sugarcane Research Institute (SRI) in Faisalabad also evaluated 60 genotypes using Principal Component Analysis (PCA) and Cluster Analysis of 19 morphological and quality traits of sugarcane (*Shahzad et al., 2016*).

To develop the sugarcane variety and evaluate its germplasm, the sugarcane fuzz of different crosses was acquired from China and grown in glasshouse conditions. The seedlings were evaluated in the field with two local check varieties (CPF-251 and HSF-240) by measuring the different morphophysiological parameters, including cane length (CL), cane diameter (CD), plant height (PH), internodal length (IL), tiller numbers (TN), brix percentage (B), chlorophyll *a* (Chl. *a*), chlorophyll *b* (Chl. *b*), and total chlorophyll (T. Chl.) content. The genetic diversity among the acquired crosses was also explored. Based on the selection criteria, we proposed some traits for variety development. We selected the top and low-performing sugarcane genotypes based on the results for further adaptability trials.

## MATERIALS AND METHODS

### Plant material and growth conditions

Sugarcane fuzz (true seed) of 78 crosses was acquired from Guangxi University, China. Sugarcane seed is tiny (approximately 1.8 mm × 0.8 mm) and tightly enclosed in bracts (*Cheavegatti-Gianotto et al., 2011*). In our experiments, seeds were not separated from the fuzz to mimic natural conditions, and the fuzz was not sterilized for the germination experiments.

### Glasshouse conditions

For sowing of the seed, trays with 50 holes were used to fill them with growth media, a mix of loamy soil and peat moss with 1:1. Seeds were sown in the glasshouse conditions having a temperature range between 28–32 °C at Day time (12–14 h) and minimum temperature was recorded between 20−22 °C at nighttime (10–12 h). Relative humidity of 60–70% is maintained constantly (*Breaux & Miller, 1987*). Watering of the trays was carried out by hand sprayer to moist the media till the germination of seedlings. Afterward, the sprinkler was used for this purpose. Seedlings transferred from trays after one and half months to separate pots for their strengthening.

### Field conditions

A field experiment was conducted at NARC, Islamabad location with Latitude: 33.6701° N and Longitude: 73.1261° E. Seedlings were transplanted after almost 60–70 days of germination in the field. The land was prepared by following the standard agronomics practices, and fertilizer application was carried out with a recommended dose of DAP and SOP at the time of sowing. Seedlings transplanted into ridges. The distance was kept at 4 ft row-to-row and plant-to-plant distance of 5 ft.

**Table 1  Pedigree information on sugarcane fuzz was used in this study.**

| Sr. No. | Genotypes | Development | Parentage |
|---|---|---|---|
| 1. | CPF-251 | Local Check | CP 87-1628 × CP 84-1198 |
| 2. | HSF-240 | Local Check | CP 43-33 (poly cross) |
| 3. | NS-1 | Fuzz | Dezhe 93-88 |
| 4. | NS-11 | Fuzz | 15 |
| 5. | NS-15 | Fuzz | Neijang 03-218 (10) |
| 6. | NS-17 | Fuzz | Guitang 02-901 X ROC 23 |
| 7. | NS-19 | Fuzz | 19 (51) |
| 8. | NS-3 | Fuzz | Funong 0708 × Funong 02-6427 |
| 9. | NS-30 | Fuzz | Chuantang 89-103 (180) |
| 10. | NS-45 | Fuzz | 169 |
| 11. | NS-46a | Fuzz | RB 72-454 × CP 94-1100 |
| 12. | NS-46b | Fuzz | RB 72-454 × CP 94-1100 |
| 13. | NS-47 | Fuzz | Yuetang 89-240 × CP 94-1100 |
| 14. | NS-4a | Fuzz | Dezhe 93-88 × Section 5 |
| 15. | NS-4b | Fuzz | Dezhe 93-88 × Section 5 |
| 16. | NS-5 | Fuzz | Funong 02-6427 |
| 17. | NS-50 | Fuzz | ROC 26 × Guitang 96-211 |
| 18. | NS-54 | Fuzz | 54 (5) |
| 19. | NS-55 | Fuzz | 05-51 × 05-407 (214) |
| 20. | NS-6 | Fuzz | Funong 0708 × Dezhe 93-88 |
| 21. | NS-63 | Fuzz | 05-51 × CP 34-425 (213) |
| 22. | NS-64 | Fuzz | CP 65-357 × ROC22 |
| 23. | NS-68 | Fuzz | Yt 91-976 × Guitang 28 |
| 24. | NS-7 | Fuzz | Guitang 05-3084 × Guitang 02-901 |
| 25. | NS-70 | Fuzz | Guitang 27 × Funong 01-0108 |
| 26. | NS-72 | Fuzz | Yt 01-71 × Funong 02-6427 |
| 27. | NS-74 | Fuzz | Yuetang 93-124 × ROC 22 |
| 28. | NS-75 | Fuzz | LCP 85-384 × ROC 22 |
| 29. | NS-77 | Fuzz | Funong 02-3924 × 07-71 |
| 30. | NS-8 | Fuzz | LCP 85-384 × Guitang 92-66 |
| 31. | NS-9 | Fuzz | HoCP 07-613 × Funong University 02-6427 |

## Phenotypic and quality parameters data collection

This study evaluated 31 sugarcane genotypes including two local check varieties, CPF-251 and HSF- 240 (Table 1). A total of six traits, including five phenotypic cane traits such as plant height (PH), cane length (CL), tiller numbers (TN), internode length (IL), and cane diameter (CD), were measured using a ruler and vernier caliper at maturity stage (10–11 months old plants). One quality parameter of cane was measured by taking the Brix% (B) data at the maturity of the plant using the handheld refractometer.

## Pigment analysis

For the measurement of Chlorophyll *a* and *b* method proposed by *Arnon (1949)* was used. Leaf samples with the same weight were immersed in ethanol overnight. Samples were

gently shacked. With the help of a spectrophotometer (Cary 60 UV-Vis; Agilent, Santa Clara, CA, USA), the concentrations were recorded at different wavelengths, 663 and 645 nm (*Adeel Zafar et al., 2021*). The following equations were used for calculations:

$$Concentration\ of\ chlorophyll\ a\left(\frac{mg}{g}F.W\right) = (12.7*A663) - (2.69*A645)$$

$$Concentration\ of\ chlorophyll\ b\left(\frac{mg}{g}F.W\right) = (22.9*A645) - (4.68*A663)$$

## Chlorophyll leaching assay

For the chlorophyll leaching assay, the method described by *Qin et al. (2011)*, was utilized. The flag leaf was harvested from individual plants, cut into small pieces (about two cm), and immersed in 30 ml of 80% ethanol at room temperature with gentle shaking in the dark. Chlorophyll quantification was carried out in tubes at 0, 50-, 100-, 150-, and 200-min. Measurements were performed in a relatively dark room with feeble light.

## Statistical analysis

The obtained phenotypic, quality, and physiological data of 29 sugarcane genotypes and two local check varieties for nine yield-related traits were subjected to descriptive statistical analysis using R software (Release 9.1.3; SAS Institute, Cary, NC, USA). Analysis of variance (ANOVA) was used to determine the variation in traits among the sugarcane genotypes (*Aguiar et al., 2018*).

The least significant difference (LSD) test-taking probability level was used to statistically separate the genotypes' mean values ($p < 0.001$). Using each trait's mean sum of squares from ANOVA results, the following genetic parameters were estimated (*Asante, Adjah & Annan-Afful, 2019*).

### Genotypic variance

$V_g = GMS - EMS$.

Here, $V_g$ is genotypic variance, $GMS$ is the grand mean square of a genotype, and $EMS$ is the error mean square.

### Phenotypic variance

$V_p = V_g + V_e$

Here, $V_p$ is phenotypic variance, $V_g$ is genotypic variance, and $V_e$ is error variance, *i.e.,* EMS.

### Heritability

$$h^2 = \frac{V_g}{/V_p} \times 100$$

Broad sense heritability (h2) was calculated in Microsoft Excel 2019 as described by (*Zaid et al., 2022*). $V_g$ is the genotypic variance, and $V_p$ is the phenotypic variance. The estimated heritability was categorized as low (0–30 percent), medium (30–60 percent), and high (>60 percent) (*Lush, 1949*).
**Table 2  Mean square values, heritability, and genetic advance for different agronomic traits of 31 (including two local checks) sugarcane genotypes.**

| SOV | DF | PH | CL | IL | TN | B | CD | Chl. *a* | Chl. *b* | T. Chl |
|---|---|---|---|---|---|---|---|---|---|---|
| Rep | 1 | $492.0^{ns}$ | $1.9^{ns}$ | $0.506^{ns}$ | $1.31^{ns}$ | $0.155^{ns}$ | $5.872^{ns}$ | $0.304^{***}$ | $0.112^{***}$ | $0.61^{***}$ |
| Geno | 30 | $2999.2^{***}$ | $2293.8^{***}$ | $13.150^{***}$ | $60.56^{***}$ | $5.078^{***}$ | $21.480^{***}$ | $12.308^{***}$ | $6.664^{***}$ | $36.72^{***}$ |
| Error | 30 | 543.3 | 213.3 | 1.721 | 0.77 | 0.826 | 2.075 | 0 | 0 | 0.04 |
| $h^2$ | | 81.89 | 90.7 | 86.91 | 98.73 | 83.73 | 90.34 | 99.91 | 98.87 | 99.52 |
| GA | | 65.32 | 63.28 | 4.6 | 11.2 | 2.75 | 6.1 | 5.11 | 3.77 | 8.82 |

**Notes.**
$^{***}$Significance at $p < 0.001$; ns, non-significance $p > 0.05$.

SOV, Source of variation; Rep, Replications; Geno, Genotypes; $h^2$, Heritability (%); GA, Genetic advance (%); DF, Degree of freedom; PH, Plant length (cm); CL, Cane length (cm); IL, Internodal length (cm); TN, Tillers number; B, Brix (%); CD, Cane diameter (mm); Chl. *a*, Chlorophyll *a* (mg/g FW); Chl. *b*, Chlorophyll *b* (mg/g FW); T. Chl, Total chlorophyll contents (mg/g FW).

### Genetic advance

$$GA = K \frac{V_g}{\sqrt{V_p}}$$

where *GA* is genetic advance, *K* is the standard selection differential, $V_g$ is the genotypic variance, and $V_p$ is the phenotypic variance, respectively. Where $K = 2.06$ at 5% selection intensity.

### Association studies

The correlation coefficient ($r$) among traits was calculated at the phenotypic level to determine the positive and negative relationship between the yield and related traits.

### Principal component analysis (PCA)

Principal component analysis (PCA) was performed to study the patterns of morphophysiological variations among 31 sugarcane genotypes including two local checks and to categorize significant variables contributing to the phenotype. PCs with eigenvalues larger than one were selected (*Jolliffe, 1972*). R 3.4.5 (*R Core Team, 2018*) was used to perform Pearson's correlation ($r$), PCA, and data visualization.

## RESULTS

To find out the significant differences among the sugarcane accessions, all the recorded data related to morpho-physiological parameters were subjected to analysis of variance (ANOVA). Results showed highly significant ($p < 0.001$) differences among the sugarcane accessions (Table 2) for all the studied traits. High broad-sense heritability was recorded for all the studied parameters, ranging from 81.89% to 99.91% (Table 2). For selecting the best genotypes, the expected genetic gain was used in addition to heritability estimates rather than just the heritability value. In this study, GA ranges from 4.6% to 65.32%. The highest GA was observed for PH (65.32%), followed by CL (63.28%, Table 2).

### Mean variability
#### Plant height (PH, cm)
Sugarcane's higher plant height resulted from improved crop growing conditions and varietal characteristics. It significantly contributes to the expansion of crop biomass. In this

**Table 3** Summary statistics of 31 (including two local checks) sugarcane accessions for different agronomics traits.

| Traits | Maximum | Minimum | Mean | Variance | Standard deviation |
|---|---|---|---|---|---|
| PH | 429.77 | 275.84 | 383.98 | 1,499.58 | 38.72 |
| CL | 283.46 | 126.49 | 209.64 | 1,146.88 | 33.87 |
| IL | 17.25 | 8.75 | 12.87 | 6.58 | 2.56 |
| TN | 31.50 | 3.50 | 10.24 | 30.28 | 5.50 |
| B | 21.10 | 14.20 | 18.23 | 2.54 | 1.59 |
| CD | 36.88 | 21.27 | 27.66 | 10.74 | 3.28 |
| Chl. *a* | 14.12 | 2.64 | 5.97 | 6.75 | 2.60 |
| Chl. *b* | 9.99 | 1.64 | 4.84 | 3.42 | 1.85 |
| T. Chl | 24.11 | 4.44 | 10.81 | 19.20 | 4.38 |

**Notes.**
PH, plant length (cm); CL, Cane length (cm); IL, Internodal length (cm); TN, Tillers number; B, Brix (%); CD, Cane diameter (mm); Chl. *a*, Chlorophyll *an* (mg/g FW); Chl. *b*, Chlorophyll *b* (mg/g FW); T. Chl, Total chlorophyll contents (mg/g FW).

study, plant height (PH) varied from 275.84 to 429.77 cm, with an overall mean of 383.98 cm (Table 3). The maximum PH was observed for CPF-251, while the smallest PH was recorded in NS-74 (Fig. 1A).

## Cane length (CL, cm)
Cane length (CL) is considered a vital trait for improving PH. Its range varied from 126.49 to 283.46 cm, with an overall mean of 209.64 cm (Table 2). Maximum and minimum CL were recorded in NS-9 (283.46) and NS-19 (126.49), respectively (Fig. 1B).

## Internodal length (IL, cm)
The performance of 31 sugarcane genotypes for internodal length (IL) was variable, ranging from 8.75 to 17.25 cm (Table 3). While the minimum IL was recorded in NS-7 (8.75 cm), followed by NS-74 (9.3 cm) and NS-19 (9.6 cm, Fig. 1C). The maximum IL was observed in NS-17 (17.25 cm), followed by NS-64 (17 cm) and NS-4b (16.5 cm).

## Tiller number (TN)
The yield of any crop depends on the number of productive tillers. The Tiller number (TN) varied from 3.50 to 31.50. Maximum TN was observed for the cultivar NS-6 (31.5), while less TN was observed in NS-54 (3.5). Overall, sugarcane genotypes showed more TN as compared to local checks (Fig. 2A).

## Brix percentage (B, %)
The juice's total solids content, expressed as a percentage, is referred to as its juice brix (B). Both sugars and non-sugars are included in brix. In this study, B varied from 14.20 to 21.10% (Table 3). Maximum B was observed in NS-5 (21.1%), while minimum B was observed in NS-4a (14.2%). One and 16 genotypes performed better for B as compared to local checks CPF-251 and HSF-240, respectively (Fig. 2B).

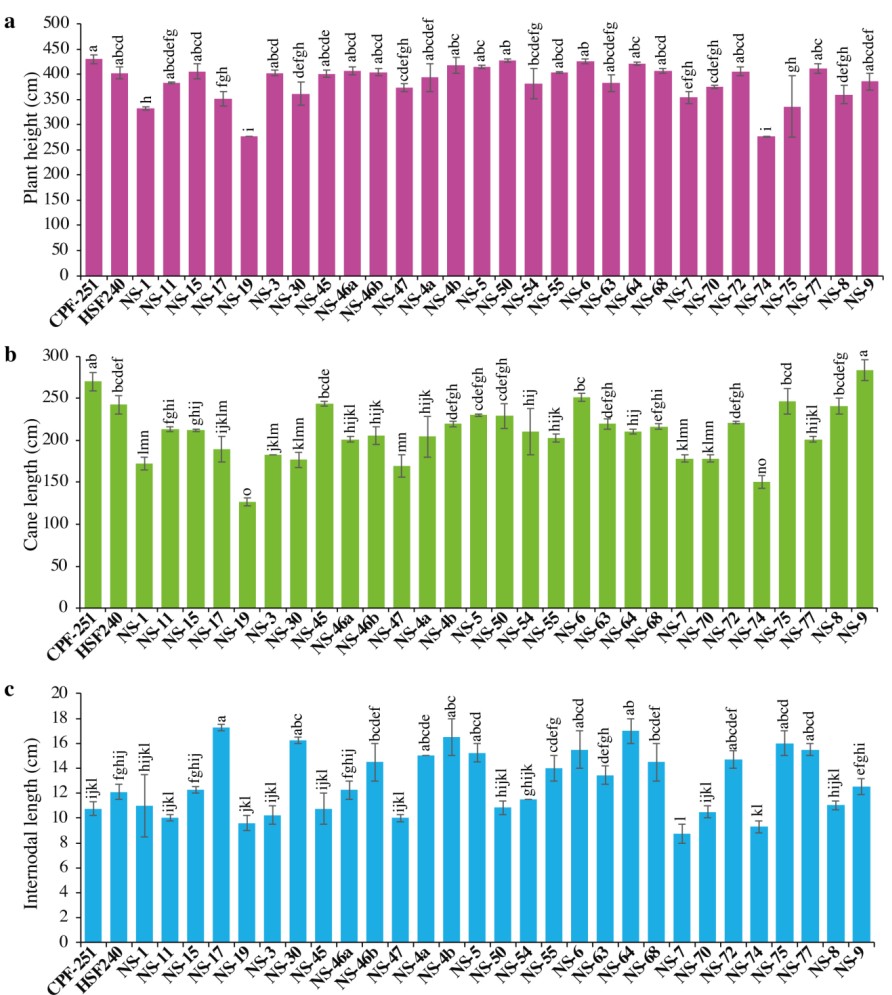

**Figure 1** **(A–C) Comparison of 31 genotypes (including two local checks) of sugarcane for plant height, cane length, and internodal length.** Different lowercase letters above the column differ significantly at $p < 0.05$. Significant levels were checked through LSD.

### Cane diameter (CD, mm)

In sugarcane, yield is influenced by many factors, including morphological characteristics like the number of tillers per plant, cane length, and diameter. The cane diameter (CD) obtained by the genotypes ranged from 21.27 to 36.88 mm (Table 3). The maximum and minimum CD was observed for the genotypes NS-46a (36.88 mm) and NS-6 (21.27 mm), respectively (Fig. 2C).

## Chlorophyll contents

Photosynthesis is a complex process that occurs in particular organs known as chloroplasts. The leaves acquire their green color from the direct absorption of blue and red sunlight by the green photosynthetic chlorophyll. In this study, NS-30 (14.12 mg/g FW) showed the highest amount of Chl *a* followed by NS-50 (10.51 mg/g FW), NS-72 (10.37 mg/g FW),

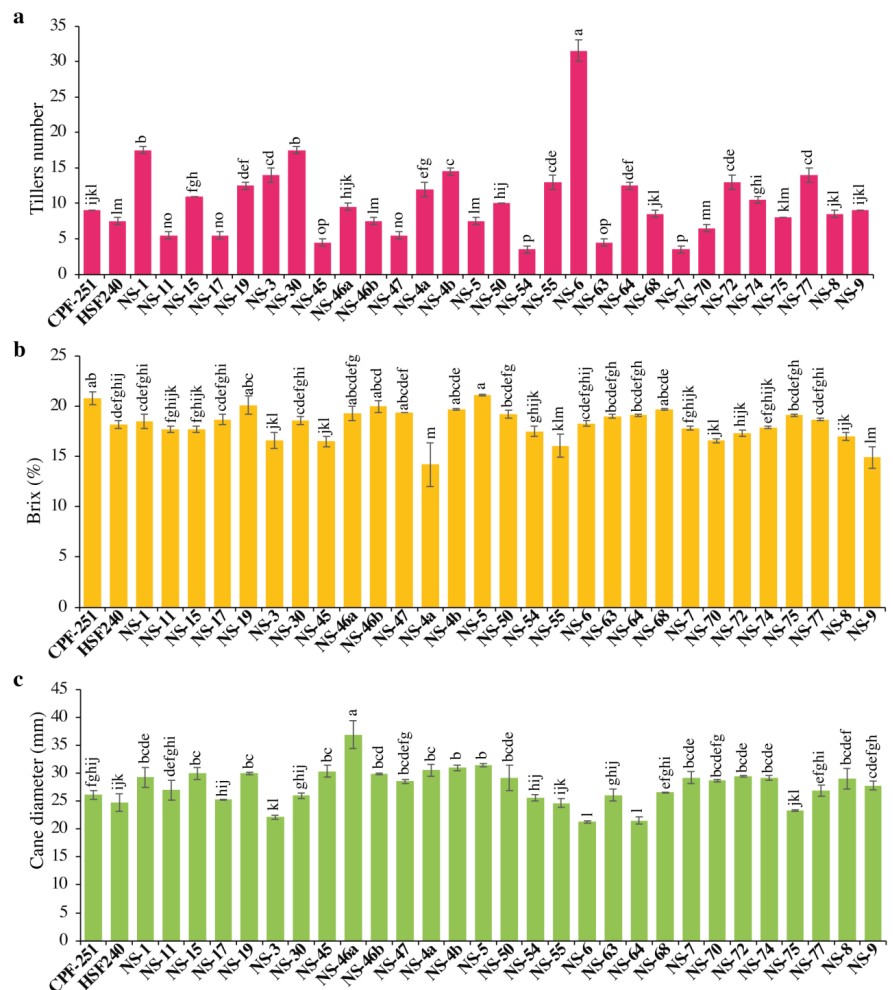

**Figure 2** (A–C) Comparison of 31 (including two local checks) genotypes of sugarcane for tiller numbers, brix percentage, and cane diameter. Different lowercase letters above the column differ significantly at $p < 0.05$. Significant levels were checked through LSD.

and NS-75 (9.90 mg/g FW), while the genotypes NS-9 (2.63 mg/g FW), NS-5 (2.79 mg/g FW), and NS-8 (2.83 mg/g FW) showed less amount of Chl *a* (Fig. 3). We also observed Chl *b* in all the sugarcane genotypes. NS-30 (9.98 mg/g FW) showed the highest amount of Chl *b*, followed by NS-72 (7.99 mg/g FW) and NS-75 (7.64 mg/g FW). Furthermore, the genotypes NS-30 (24.11 mg/g FW) and NS-72 (18.36 mg/g FW) showed higher T. Chl, while the genotypes NS-5 (4.44 mg/g FW) and NS-6 (5.73 mg/g FW) have a minimum amount of T. Chl (Fig. 3).

## Chlorophyll leaching assay

A chlorophyll leaching test was carried out to check the cuticular membrane permeability in all the mature leaves of 31 sugarcane genotypes (including two local checks). For this purpose, we checked the leaching at different time points (0, 50, 100, 150, and 200 min).
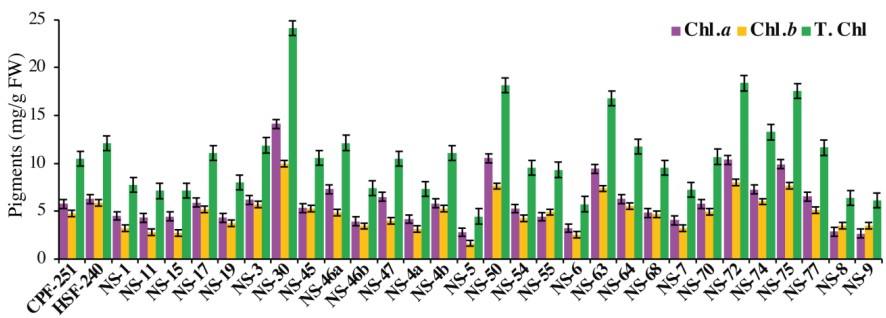

**Figure 3** Measurement of chlorophyll pigments in 31 genotypes (including two local checks) of sugarcane.

A significantly higher amount of chlorophyll contents was extracted from the genotypes NS-30, NS-50, NS-63, NS-72, and NS-75 compared to the genotypes NS-5, NS-6, NS-9, NS-8, and NS-15 at the time point of 200 min (Fig. 4). An almost similar trend was observed for all the time points. These results showed a higher permeability rate in the earlier mentioned sugarcane genotypes, indicating the poor cuticle developments in the said genotypes.

## Evaluation of genetic diversity through Principal Component Analysis

Principal component analysis (PCA) was previously used to estimate genetic diversity in different crops (*Khan & Anwar, 2019*). In PCA, a dataset is considered consultant, while the cumulative percentage of the variance of the essential additives is more than 80%. In this study, we also performed the PCA to check the variability and association relations of agronomic traits among the sugarcane genotypes (Fig. 5). Sugarcane genotypes were distributed in biplots by group, with traits represented by different vectors. The cumulative variance of the first eight PCs was 99.9%, and the variability among the PCs was 0.5 to 34.4% (Fig. 5A). The first three PCs (PC1 = 34.4%, PC2 = 22.5%, and PC3 = 13.1%) were significant. Chl *a*, Chl *b*, and total chlorophyll contents were the major contributing factors in PC1. Similarly, in the PC2 PH, CL, and IL were the main factors, while in the PC3 TN, TD, CL, and PH showed the highest loading values (Fig. 5B). Projection of traits showed that CL and IL strongly related to PH. Similarly, the traits Chl *a*, and Chl *b* are related to T. Chl (Fig. 5C). The genotypes NS-5, NS-9, NS-4a, and NS-15 were opposite to the NS-30, NS-50, NS-63, and NS-75 (Fig. 5C). Based on PCA results, these genotypes can be categorized as best and poor performers and can be utilized in future breeding programs.

## Association among the morpho-physiological attributes of sugarcane

Association among any two traits is known as correlation. Pearson's correlation results of this study showed that variation is present among the parameters. The traits such as PH, CL, and IL have strong associations. Similarly, PH showed a positive association with CL and IL, while it showed a negative association with the chlorophyll pigments. Chl *a* and
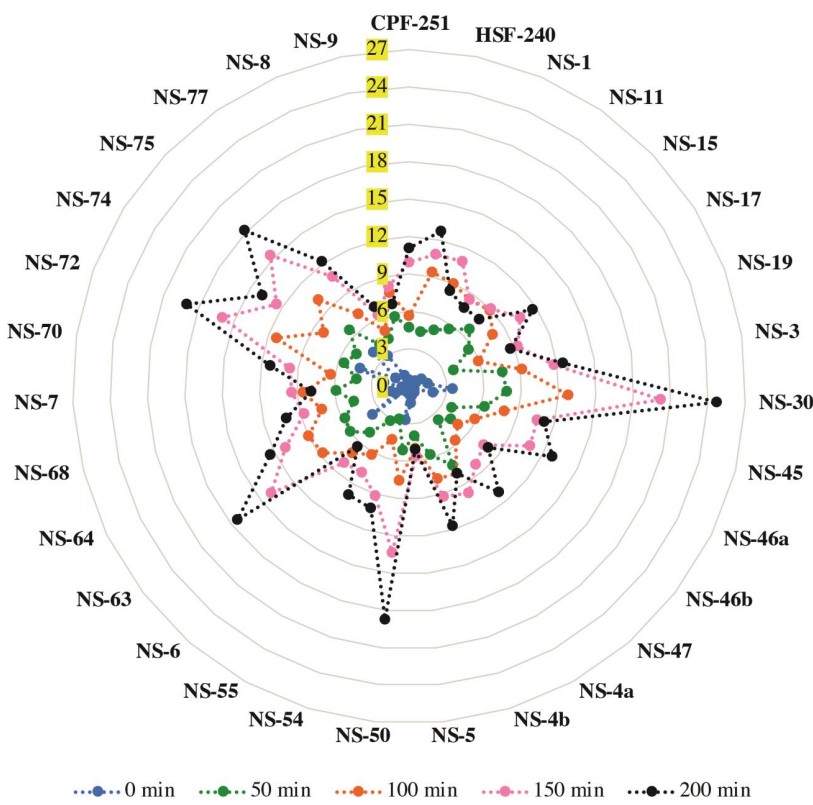

**Figure 4 Chlorophyll leaching assay of 31 (including two local checks) sugarcane genotypes.** Samples were shaken on the shaker, and readings were taken at different time points and labeled with different colors. Values are the average of three biological replicates.

Chl *b* have potent associations with T. Chl. Cane diameter negatively correlated with all the studied parameters (Fig. 6).

## Cluster analysis of sugarcane accessions by using morpho- physiological attributes

The fuzz of exotic sugarcane crosses and two local checks (CPF-251 and HSF-240) were grown in the glasshouse and shifted to the field for further evaluation (Fig. 7). To study the diversity among the sugarcane accessions, hierarchical cluster analysis (HCA) was used by using recorded morphological data from the mentioned sugarcane genotypes. Such examination gives a superior circulation of accessions under various conditions as well as various gatherings of accessions in a similar climate. This method generated a phylogenetic tree based on the morpho-physiological attributes of sugarcane, on which the 31 sugarcane accessions including two local checks were distributed into four distinct clusters (similarity coefficient threshold = 25). These clusters were shown with different colors and named "tolerant (T, purple), moderate tolerant (MT, green), moderately susceptible (MS, red), and susceptible (S, blue). Additionally, these clusters were further divided into two distinct subgroups on the basis of similarities among the sugarcane accessions (Fig. 8).

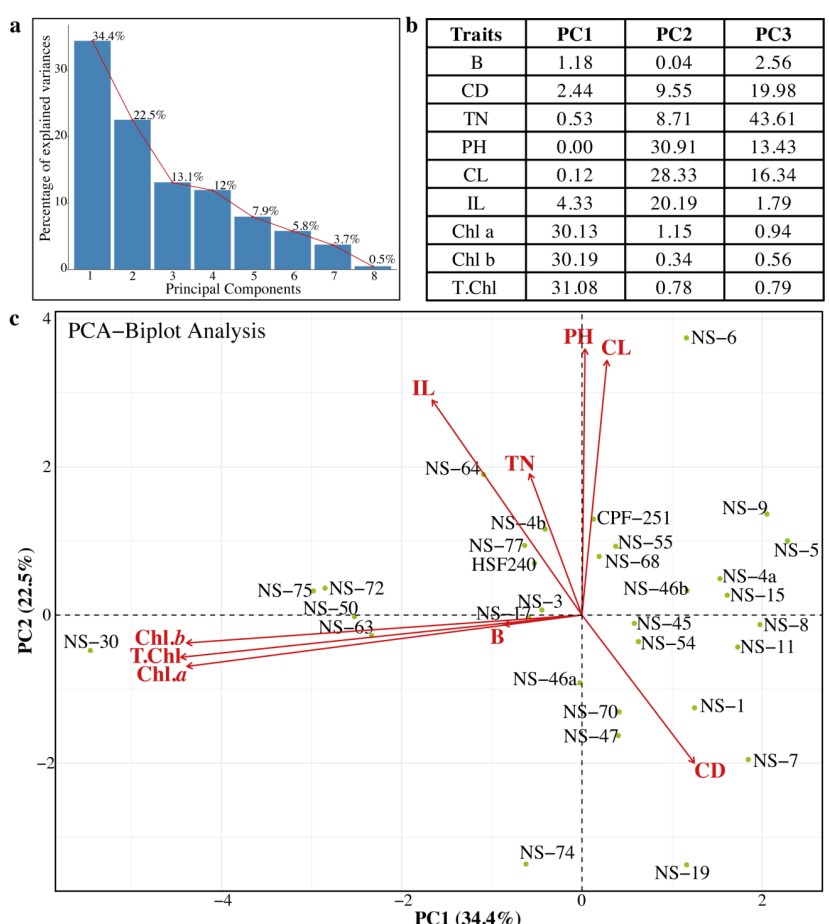

**Figure 5** (A–C) Principal component analysis (PCA) among the different parameters of sugarcane. PH, plant length (cm); CL, Cane length (cm); IL, Internodal length (cm); TN, Tillers number; B, Brix (%); CD, Cane diameter (mm); Chl. *a*, Chlorophyll *a* (mg/g FW); Chl. *b*, Chlorophyll *b* (mg/g FW); T. Chl, total chlorophyll contents (mg/g FW).

## DISCUSSION

Sugarcane is an economically important crop because it provides food, feedstock, and biofuel (*e.g.*, ethanol), which can address global food and energy challenges, particularly in developing countries. Furthermore, sugarcane is important for the environment because it is a C4 plant with high photosynthetic efficiency and requires less water and nitrogen compared to other crops. It can also sequester carbon in the soil, making it an important tool for mitigating climate change (*Budeguer et al., 2021*; *de Souza Oliveira et al., 2021*). Sugarcane production in Pakistan is hindered by several biotic and abiotic factors, including temperature, humidity, lack of technology implementation, a complicated genome, a low fertility rate, a long selection of breeding cycles, an unfavorable day length during the flowering period, and a poor seed setting rate in ordinary environments (*Sanghera et al., 2019*). Pakistan lacks the essential facilities for cultivating hybrid seeds, planting approved and high-yielding varieties, and genomic improvement in sugarcane varieties (*Tiawari*

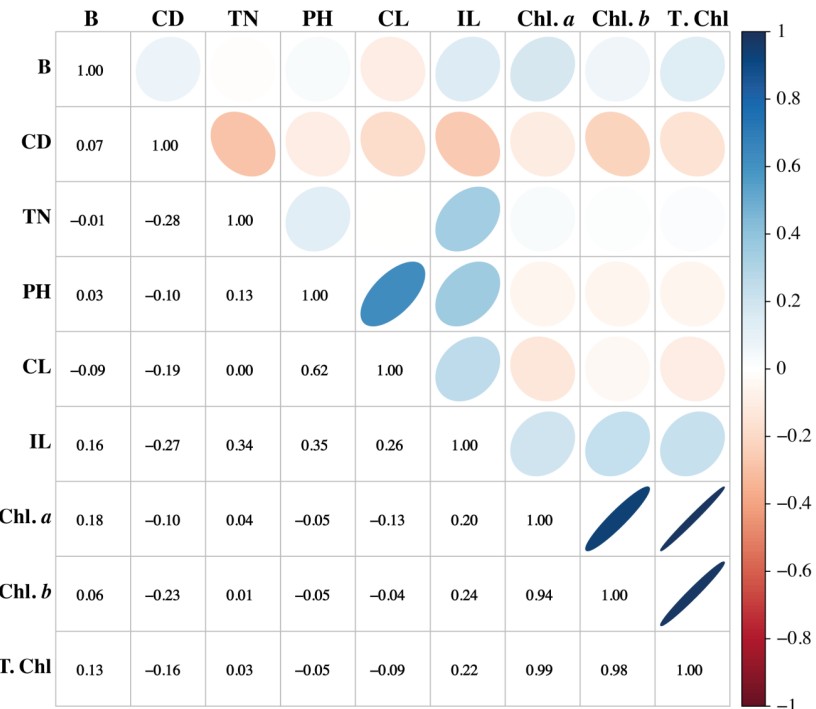

**Figure 6  Association among the morpho-physiological parameters of sugarcane.** B, Brix (%); CD, Cane diameter (mm); TN, Tillers number; PH, Plant length (cm); CL, Cane length (cm); IL, Internodal length (cm); Chl. *a*, Chlorophyll *a* (mg/g FW); Chl. *b*, Chlorophyll *b* (mg/g FW); T. Chl, total chlorophyll contents (mg/g FW).

*et al., 2009*). Therefore, the sugarcane breeding research institutes in Pakistan, including SRI-AARI, Faisalabad; Sugar Crops Research Institute, Mardan; and NSTHRI, Thatta, mainly import sugarcane fuzz or new varieties from different countries, such as China, USA, and Brazil.

Previous studies have evaluated sugarcane crops only in field conditions. In this study, we germinated the imported fuzz in glasshouse conditions and later shifted the seedlings to the field. We optimized the protocol and other requirements of the germination. The recorded data showed variability for all the parameters. The population's genetic variation was large, determining whether a program to improve variety would be successful. Parameters with high heritability and genetic advance can be used as a selection criterion for the sugarcane improvement (*Kumar & Rajamani, 2004*). Chloroplasts are the main cell organelles for photosynthesis. Chlorophyll is a substance in chloroplast that reflects green light and is used to measure productivity, stress, and photosynthetic capacity (*Uddling et al., 2007*). Chlorophyll concentration in the leaf is an indicator of chloroplast development, it's capacity for photosynthetic activity, the level of leaf nitrogen, and overall plant health. Plant nitrogen content is known to be almost directly proportional to its chlorophyll content. In this study, the chlorophyll contents (*a*, *b*, and total) of the genotypes were significantly different.

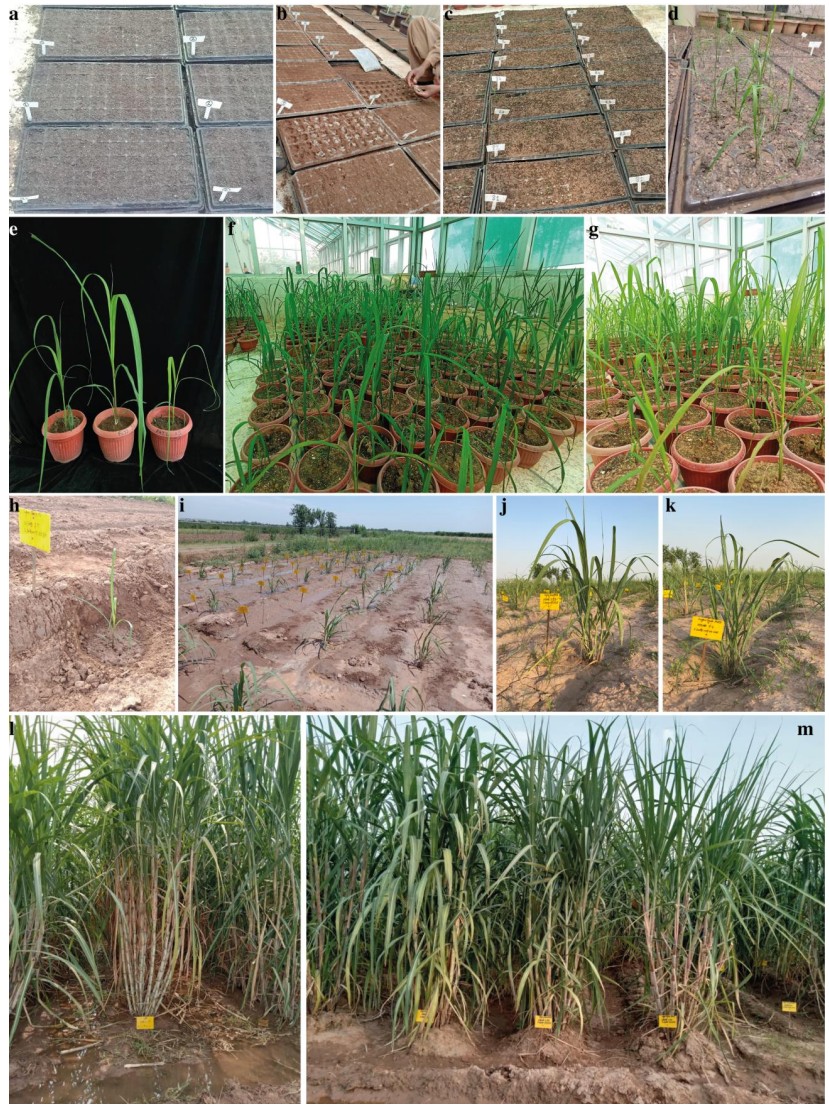

**Figure 7  Different growth stages of sugarcane.** (A) Preparation of sowing trays. (B) Manually sowing of sugarcane fuzz. (C–D) Emergence and germination of young seedlings from the fuzz. (E–G) Germinated seedlings were shifted into pots. (H–I) Shifting and establishment of sugarcane seedlings into the field. (J–K) Tillering and canopy development stages of sugarcane. (L–M) Elongation and maturing phase of sugarcane.

Cane yield is a complicated phenomenon that depends on its components and their interactions with each other. To improve any trait in a breeding program, it is necessary to learn how it relates to other traits. In a crop breeding program, it is helpful to understand the relative importance of various plant traits (*Uzair et al., 2022b*). Correlation analysis can help in developing a directional model for yield prediction and provide information on the associated responses of plant characters. PH and CL are direct factors that control the cane yield (*Portz, Amaral & Molin, 2012*; *Shukla et al., 2017*; *Tena, Mekbib & Ayana, 2016*)

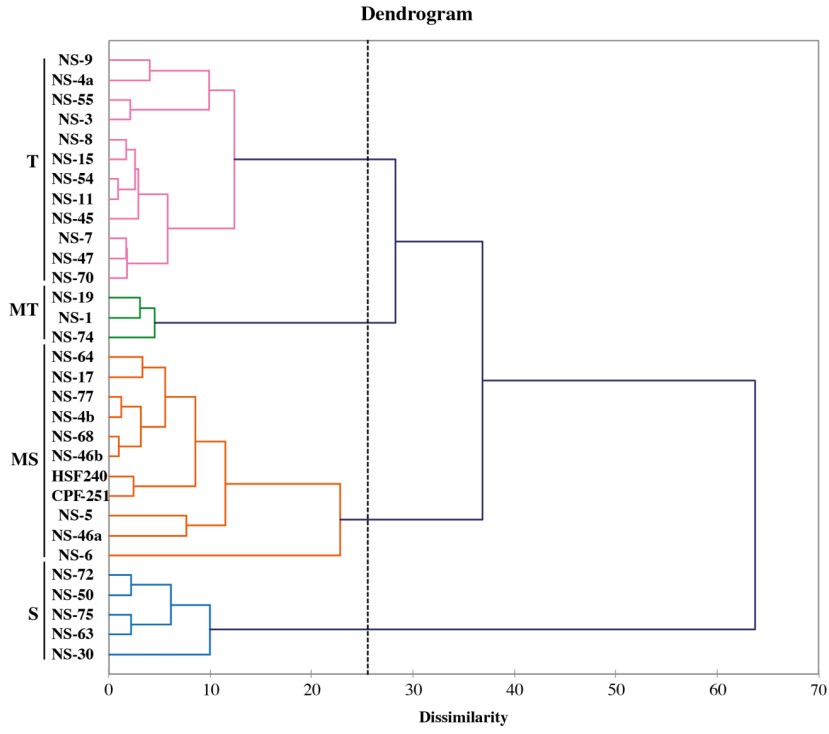

**Figure 8  Classification of sugarcane genotypes on the bases of their performance.** Genotypes were grouped into four groups susceptible (S, blue), moderately susceptible (MS, red), moderately tolerant (MT, green), and tolerant (T, purple).

and can be improved by increasing the IL of the sugarcane plants because of a positive correlation between the three parameters. Our results were supported by those of such studies.

Clustering techniques, such as PCA and partial least squares-discriminant analysis are commonly employed to investigate the variations among samples of various groups and determine the primary parameters (*Azam et al., 2020*; *Deng et al., 2020*; *Uzair et al., 2022a*; *Uzair et al., 2022b*). A biplot analysis is primarily used to determine variables that can be categorized into fundamental groups and subgroups based on homogeneity and uniqueness for parental choice in breeding programs. In this study, PCA on nine morphophysiological parameters showed that the first three PCs captured 70% of the total variations that primarily impacted PH, CL, IL, and TN, making them indicators for selection. PCA results showed that NS-4a, NS-5, NS-6, NS-9, NS-8, and NS-15 were better performers, while NS-30, NS-50, NS-72, and NS-75 were poor performers.

## CONCLUSION

A total of 31 sugarcane genotypes (29 exotic crosses and two local checks) were evaluated for genetic diversity and showed significant variations for all the studied parameters. Based on these findings, we conclude that PH, CL, IL, and TN parameters must be considered

when selecting of sugarcane genotypes for higher cane production. Genotypes such as NS-4a, NS-5, NS-6, NS-9, NS-8, and NS-15 were found to perform better than others. This study provides valuable insights into the genetic diversity and performance of sugarcane genotypes, and the findings could be useful in future breeding programs for improving sugarcane production and benefiting stakeholders and farmers in sugarcane-producing areas.

### Funding

This research was funded by Researchers Supporting Project number (RSP-2023R369), King Saud University, Riyadh, Saudi Arabia. The PSDP and the Ministry of National Food Security and Research (MNFS&R) provided resources for this study. Sugarcane Research Institute, Faisalabad provided local checks. The Guangxi Key Lab for Sugarcane Biology, Nanning, China, provided the sugarcane fuzz. This study was conducted from the funds of the Ministry of National Food Security and Research of Pakistan released for the PSDP-project (760) ''Sino-Pak Agricultural Breeding Innovations Project for Rapid Yield Enhancement. The funders had no role in study design, data collection and analysis, decision to publish, or preparation of the manuscript.

### Grant Disclosures

The following grant information was disclosed by the authors:
Researchers Supporting Project:  RSP-2023R369.
King Saud University, Riyadh, Saudi Arabia.
The PSDP and the Ministry of National Food Security and Research (MNFS&R).
Sugarcane Research Institute, Faisalabad.
The Guangxi Key Lab for Sugarcane Biology, Nanning, China.
Ministry of National Food Security and Research of Pakistan.
Sino-Pak Agricultural Breeding Innovations Project.

### Competing Interests

Sajid Fiaz is an Academic Editor for PeerJ.

### Author Contributions

- Bilal Saleem conceived and designed the experiments, performed the experiments, prepared figures and/or tables, authored or reviewed drafts of the article, and approved the final draft.
- Muhammad Uzair conceived and designed the experiments, performed the experiments, analyzed the data, authored or reviewed drafts of the article, and approved the final draft.
- Muhammad Noman performed the experiments, authored or reviewed drafts of the article, and approved the final draft.
- Kotb A. Attia analyzed the data, prepared figures and/or tables, and approved the final draft.

- Muqing Zhang conceived and designed the experiments, authored or reviewed drafts of the article, and approved the final draft.
- Mona S. Alwahaibi analyzed the data, prepared figures and/or tables, authored or reviewed drafts of the article, and approved the final draft.
- Nageen Zahra performed the experiments, authored or reviewed drafts of the article, and approved the final draft.
- Muhammad Kashif Naeem analyzed the data, prepared figures and/or tables, and approved the final draft.
- Arif A. Mohammed analyzed the data, authored or reviewed drafts of the article, and approved the final draft.
- Sajid Fiaz conceived and designed the experiments, prepared figures and/or tables, and approved the final draft.
- Itoh Kimiko analyzed the data, prepared figures and/or tables, and approved the final draft.
- Muhammad Ramzan Khan conceived and designed the experiments, analyzed the data, authored or reviewed drafts of the article, and approved the final draft.

### Ethics

The following information was supplied relating to ethical approvals (i.e., approving body and any reference numbers):

National Institute for Genomics and Advanced Biotechnology (NIGAB), National Agricultural Research Centre, Park Road, Islamabad, Pakistan

### Data Availability

The raw data is available in the Supplementary File.

### Supplemental Information

Supplemental information for this article can be found online at http://dx.doi.org/10.7717/peerj.15646#supplemental-information.

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
