# Peer review of "Evaluation of sugarcane promising clones based on the morphophysiological traits developed from fuzz"

_PeerJ, doi:10.7717/peerj.15646_

## Round 0.1 · original submission · Major Revisions

This is an interesting article and needs to be published, the next step is the revisions of various sections of the manuscript which need more elaboration keeping in view the comments from the esteemed reviewers. In addition, some sections are deficient in background and hypothesis, lack consistency/story, and have incomplete sentences. I am seeing citations in the results sections.

Reviewer 1 ·

Basic reporting

.

Experimental design

.

Validity of the findings

.

Additional comments

In this manuscript, the authors reported a very important topic related to sugarcane. However, I have the following concerns:
1. Enlist some recent studies that used these parameters for varieties evaluation and development.
2. The author has mentioned why Pakistan import the seed of sugarcane from other countries, give details and reference.
3. Does the authors have information quantity of Fuzz being imported from various countries in Pakistan?
4. The author has mentioned the growing conditions in a glass house, provide a suitable reference.
5. At which stage the phenotypic parameters were collected, please specify the time after the date of sowing.
6. You have used the chlorophyll leaching assay that has a role in different stress conditions, please elaborate on the background/mechanism of using this method and add more previous studies related to sugarcane if present.
7. Provide the genetic background of the crosses used in this study.
8. Discussion need improvement, present for is much shallow.
9. Does the pedigree information is present, if yes author need to include in tabulated form, within MM section of the article.
10. Carefully check and remove spelling and other grammatical mistakes.

Reviewer 2 ·

Basic reporting

The article is well written. It has sufficient background and well structured

Experimental design

The study fits with scope of the journal. The research question is will defined and methods are well described.

Validity of the findings

Work is interesting and has novelty. Data to support authors hypothesis are provided.

Additional comments

I have reviewed the article and found it interesting with larger emphasis given on breeding side of sugarcane. Sugarcane is an important crop, and varietal development always remain demanding for farming and industrial sector.
I have some comments which need to be address in the revised version of the article before getting my support to endorse article for publication.
1. Abstract: The abstract does not need to have detailed background information, there must be more emphasis on novel results, discussion and future prespective.
2. Keywords: Avoid the keywords given in the title of the article.
3. Introduction: Author need to add some information on genetics side, so it gave knowledge on genetics aspect to the readership community.
4. Be consistent with the abbreviations, the full form need to mentioned at first place, abbreviation can be utilized subsequently.
5. The outdated citations need to be removed, a research article may be 35-40 citations.
6. The objectives of the study are not clear, what I understand is the whole study was conducted to check low and high performing genotypes.
7. Material and Methods: there is disparity of genotypes used to conduct experiment. At one place author have mentioned 31 genotypes and two check varieties whereas, at another place mentioned 29 genotypes and two check varieties.
8. Results: There are various overlapping and confusing statements such as line 180-182.
9. Line 293-294, which two distinct groups, kindly write a bit detail.
10. Discussion: sugarcane is a critical crop, how?
11. Line 305-306, it would be better if you mentioned the experimental stations from Pakistan import fuzz (true seed).
12. Line 307-311 is overlapping statement mentioned above in MM section.
13. Overall, discussion is weak need good effort from authors to make considerable improvement.
14. Conclusion section is very generic, make improvement.
15. Figure 8 need to be moved to result section.

Reviewer 3 ·

Basic reporting

.

Experimental design

.

Validity of the findings

.

Additional comments

I have reviewed the article, it’s an interesting study with practical applications. I have few comments/suggestions for the authors to consider during revision of the article.
1. Abstract is very generic, need effort to make it scientific.
2. Introduction lack with basic information on genetics, evolution and other important information, the objectives are not well described.
3. Authors are not consistent with the abbreviations.
4. Remove outdated citations/references.
5. Growth condition method was adopted by any other researcher or authors developed by own.
6. The SI units, symbols and equation must be on standard format.
7. Remove citations from the result section of the article.
8. How pH can be shortest? Author has mentioned in the results section of the article.
9. Some of the results are overlapping, make sure to keep the results streamlined.
10. Discussion is weak, author need to make substantial effort to improve.
11. Authors need to improve the scientific language of the article.

---

## Round 0.2 · Minor Revisions

I suggest to the authors for reading all of sections carefully for removal of English mistakes/grammar/errors by using the track changes option in MS office, then submit within due dates.

---

## Round 0.3 · Minor Revisions

I'm unable to see point to point response from the authors. The attached email about response to editors is not acceptable in current form.

By the way, why the names of softwares and citations were not mentioned in the manuscript, further very recent literature (2023) published in from peer-reviewed journals.

In addition, the experimental conditions, genetic background of accessions, and site of experiment with reference to GPRS is missing in the revised version.
I'm proposing these comments because these are very basic parts of any manuscript.

I suggest that authors must see English grammar/errors in all the sections critically before submission of the revised version.

---

## Round 0.4 · accepted · Accept

I found the changes according to the comments from reviewers.